# Experiences of hospice dementia care: A qualitative study of bereaved carers and hospice clinicians

A. Bosco[1,2,3]*, C. Di Lorito[4], M. Dunlop[1], A. Booth[1], D. Alexander[5], S. Jones[5], B. R. Underwood[6,7], C. Todd[1,3,8,9,10], A. Burns[8,9,11]

1 School of Health Sciences, Faculty of Biology, Medicine and Health, The University of Manchester, Manchester, United Kingdom, 2 Primary Care Unit, University of Cambridge, Cambridge, United Kingdom, 3 National Institute for Health and Care Research, Applied Research Collaboration- Greater Manchester, Manchester, United Kingdom, 4 Department of Primary Care and Population Health, Centre for Ageing Population Studies, Royal Free Hospital, University College London, London, United Kingdom, 5 East Cheshire Hospice, Macclesfield, Cheshire, United Kingdom, 6 Department of Psychiatry, University of Cambridge, Cambridge, United Kingdom, 7 Cambridgeshire and Peterborough NHS Foundation Trust, Windsor Unit, Fulbourn Hospital, Cambridge, United Kingdom, 8 Manchester Institute for Collaborative Research on Ageing, Manchester, United Kingdom, 9 Manchester Academic Health Science Centre, Manchester, United Kingdom, 10 Manchester University NHS Foundation Trust, Manchester, United Kingdom, 11 Division of Neuroscience and Experimental Psychology, The University of Manchester, Manchester, United Kingdom

* Alessandro.bosco@manchester.ac.uk

**Data Availability Statement:** Data contain potentially sensitive information as only one hospice acted as recruitment site. Data are stored by a third-party organization (University of

## Abstract

### Background

Nearly 50 million people worldwide have dementia and the increasing numbers requiring end-of-life and palliative care, has led to national efforts to define standards of care for this patient group. Little research, however, has been done to date about the experience of hospice care for people with dementia accessing these services. This study explores the views of hospice dementia care for bereaved carers of people with dementia and hospice clinicians.

### Methods

We used purposive sampling for participant recruitment. Semi-structured qualitative interviews were conducted with bereaved carers and hospice clinical staff. Interviews were audio recorded and the transcriptions were analysed through thematic analysis. A total of 12 participants were interviewed from one service in the Northwest region in the UK. All were female and white British.

### Results

Participants described their experience of hospice dementia care in three main themes: Pre-access to service, roles and responsibility within hospice care, ease and difficulty of last period of end-of-life care.

Manchester). Data requests can be submitted to the relevant REC committee (Protocol number: NHS001995; reference: 22/SC/0362), Email: oxfordc.rec@hra.nhs.uk.

**Funding:** This paper presents independent research funded by East Cheshire Hospice and [in part] by the National Institute for Health and Care Research Applied Research Collaboration-Greater Manchester (NIHR200174) and the National Institute for Health and Care Research by a Senior Investigator Award to Prof Todd (NIHR200299). The views expressed are those of the authors and not necessarily those of the East Cheshire Hospice, NHS, the NIHR, the Department of Health and Social Care, or its partner organisations. BRU's post is part funded by a generous donation from Gnodde Goldman Sachs Giving. The specific roles of these authors are articulated in the 'author contributions' section. The funders had no role in study design, data collection and analysis, decision to publish, or preparation of the manuscript.

**Competing interests:** The authors have read the journal's policy and have the following competing interests: BRU received funding from Gnodde Goldman Sachs Giving. This does not alter our adherence to PLOS ONE policies on sharing data and materials.

## Conclusion

Rapid response teams delivering hospice home care could represent a better option to inpatient care and may be preferred by patients. This type of service, however, may require joined-up care with other community services, and this type of care needs to be considered and planned. Future studies should evaluate this type of community care.

## Background

Nearly 50 million individuals have dementia worldwide and estimates point at 5–8% of older people aged 60 and over have dementia [1]. Dementia is one of the major causes of disability and increased dependency in older people [1]. There is increasing awareness that dementia as a progressive condition requires appropriate end-of-life care. People with dementia, however, have poorer access to palliative care than, for example, people with cancer, and experience reduced quality of care and poorer outcomes in terms of pain management and symptoms control [2]. This has led to international collaborative work to find examples of optimal palliative care for people with dementia [3].

Person centred Care (PCC) provides a framework for promoting user engagement in healthcare decision making by sustaining the personhood of the individual with dementia in the context of their needs, values and beliefs [4]. There is consensus around the benefits of PCC initiatives for health care staff and family carers of people with dementia in hospital [5], long-term care [6] and hospice care [7]. The difference across care settings lies on the type of support they provide (e.g., personal care in residential care homes or more specialist support provided by nurses in nursing homes, or still specialist palliative care for people near the end of life in hospice settings or through hospice-at-home care options) [8].

Because of their historical role in pioneering person-centred palliative care, hospices may be optimal settings to develop new end-of-life care strategies for people with dementia [9]. A guide issued by Hospice UK [10], Hospice enabled dementia care—the first steps, is an example of PCC support provided in hospice setting in the UK for people with the condition.

Providing PCC at the end of life for people with dementia can be difficult to attain, especially in cases where other conditions or symptoms coexist [11]. Episodes of delirium in patients with dementia may require prompt and effective interventions for screening, assessment and management [12]. Although evidence suggests that PCC approaches represent best practice in end-of-life care, barriers may negatively impact on the quality of care for people with dementia. These barriers may include healthcare professionals' and carers' awareness of the progressive and terminal nature of dementia and uncertainty on illness trajectory, which make it difficult to plan care ahead of time. Barriers may ultimately lead to inappropriate and reactive care strategies [13, 14].

Advanced stages of dementia can lead to more intense and frequent behavioural and psychological symptoms that may be difficult to manage in the home environment. Advanced stages are also accompanied by increased frailty in the person who may require more specialty support in emergency situations (e.g., injuries). The first care option in these situations seemed to remain hospital A&E settings. Whilst these may be regarded as optimal care options in urgent/emergency needs, admissions to these settings in end-of-life often represent the final stay for people with dementia, who may experience death in a hospital bed and outside the familiar setting and or more adequate specialist setting (i.e. inpatient hospice or specialist dementia facility bed). When proper advanced care planning is not in place, a series of reactive

strategies may worsen care experience and lead to unfavourable consequences in the person which they may not have chosen [15].

There is currently little evidence around the care provision in hospices across the UK and care experience for health care professionals, family carers and people living with dementia or how hospices might fit in with other services providing end-of-life care for people with dementia.

Therefore, our research question was: What is the experience of end-of-life dementia hospice care for bereaved carers and hospice clinicians?

The study aimed to explore the experience of hospice dementia care for family carers and hospice clinical personnel.

## Methods

This was a qualitative study conducted over 12 months between April 2022 and March 2023. The study received ethical approval from the Health Research Authority and the South Central —Oxford C Research Ethics Committee on 18.10.2022 (REC reference: 22/SC/0362; protocol number: NHS001995; IRAS project ID: 316546).

The study complied with the AGREE Reporting Checklist for clinical practice [16] and comprised qualitative interviews with bereaved carers of people with dementia and hospice clinical staff. A logic model was used to explain the mechanisms of care delivery in end-of-life dementia care, this was informed by the Medical Research Council guidelines [17] for delivering complex interventions. Due to previously established networking, the East Cheshire region was the unit of analysis. Two members of the Patient and Public Involvement and Engagement (PPIE) group provided expertise by experience in palliative dementia care specifically for hospice settings. The PPIE members helped develop study documents (Participant information sheets, consent forms, study protocol, interview topic guides) and with study dissemination. More information on study methods and procedures for recruitment is reported in the study protocol which was previously registered on Open Science Framework (https://osf.io/pj5hq).

### Description of the hospice

In the presence of advanced dementia stages and/or frailty, the care environment may become difficult to manage, and when the person with dementia and their family are aware of alternative care options for end-of-life, more specialist hospice care is usually sought. At times, also a discharge from emergency hospital care may lead to hospice care. At the East Cheshire Hospice, outpatient service usually provides advanced care planning for people attending the Sunflower care centre (a day centre providing quality time for carers and patients engaging in group activities). East Cheshire care model entails two care options for their patients who may choose between inpatient and hospice-at-home (H@H) rapid response. Whilst the former offers 24-hour specialist support, the latter allows the care to be delivered directly at the home of the person, with little disruption in the familiar environment. H@R rapid response clinicians would co-ordinate the care with primary and secondary community services to ensure that out-of-hour (OOH) support is provided effectively to the person with dementia. Care co-ordination may prove more challenging to plan as multiple healthcare and social services need to be in place, therefore, patients with more complex cases are advised to receive inpatient care which provides 24-hour support. This would reduce the burden of care for carers who may become the first port of call for distressful situation during OOH.

Patients choosing inpatient care can still benefit from a friendly environment, as allowed to personalise their rooms, however more work is currently being done to make the facility more dementia friendly. Carers of either outpatient or inpatient care can benefit from bereavement

support with the hospice clinical team scheduling a follow-up to promote their emotional and mental wellbeing. Family carers whose loved ones stayed without specialist hospice care support (e.g., people remaining in their home without specialist end-of-life care or those admitted to hospital inpatient care) may be at increased risk of not receiving adequate bereavement support.

Discharge may occur from hospice inpatient care when the person's stay is longer than expected (usually over a month) and or when the person experiences a remission of symptoms, and no end-of-life care is needed. These may be times when the uncertainty of dementia may pose both the person with dementia and their carers at increased fear of losing support from hospice clinicians. From the available cases in our study, also carers who were informed about the possibility of discharge reported that it was difficult to accept, as the fear of a transition in caring role and becoming full time carers in complex care situations was greater than having the person released. The East Cheshire end-of-life and palliative care pathway for dementia is reported in Fig 1.

## Study procedure

We followed a similar strategy for the recruitment of bereaved carers and clinical hospice personnel who were recruited from 12/12/2022 to 27/01/2023. For hospice clinical personnel, the team leader in the participating hospice discussed at team meetings whether the staff would be happy to be contacted by the research team to explore volunteer involvement in the study as participants. For bereaved carers, the clinical hospice lead identified potential carers against study eligibility criteria. Eligible carers were contacted by the clinical hospice lead and were provided with details of the study. They were asked whether they would be happy to be contacted by the research team to further discuss study involvement. All participants who accepted to take part in the study, were required to sign a consent to contact form that allowed the team leader in the participating hospice to share their details with the research team. All participants were given an information sheet and those who agreed to take part in the qualitative interviews were required to sign a consent form before the interview could take place. At the end of the interviews, they were provided with a debriefing form summarising the aims of the study and how to address any issues (if any) that could occur during or after the study.

Bereaved carers were only approached if a minimum of 4 weeks had elapsed since the passing away of their loved one with dementia.

## Inclusion criteria

Carers were eligible if they were English speaker, a bereaved family carer of a person with a formal diagnosis of dementia (either spouse or children) who accessed hospice care for their end-of-life disease trajectory. Carers were eligible if the person they cared for had passed away more than one month before study recruitment (to reduce the distress of recalling emotional experiences of care) and within the previous year (this was to ensure an easier retrieval of information). Carers were further eligible if they were willing to discuss about dementia and their caring experiences. There was no restriction on gender.

Hospice clinical personnel were eligible if they were 25 years or over (an age by which basic clinical qualifications would be completed), English speaker, working in hospice and with people living with dementia, if they were willing to discuss about their experiences. There was no restriction on gender.

## Data collection

Semi-structured, qualitative interviews were held over the phone with staff and bereaved relatives about dementia care at the end of life. We used purposive sampling to find a defined and

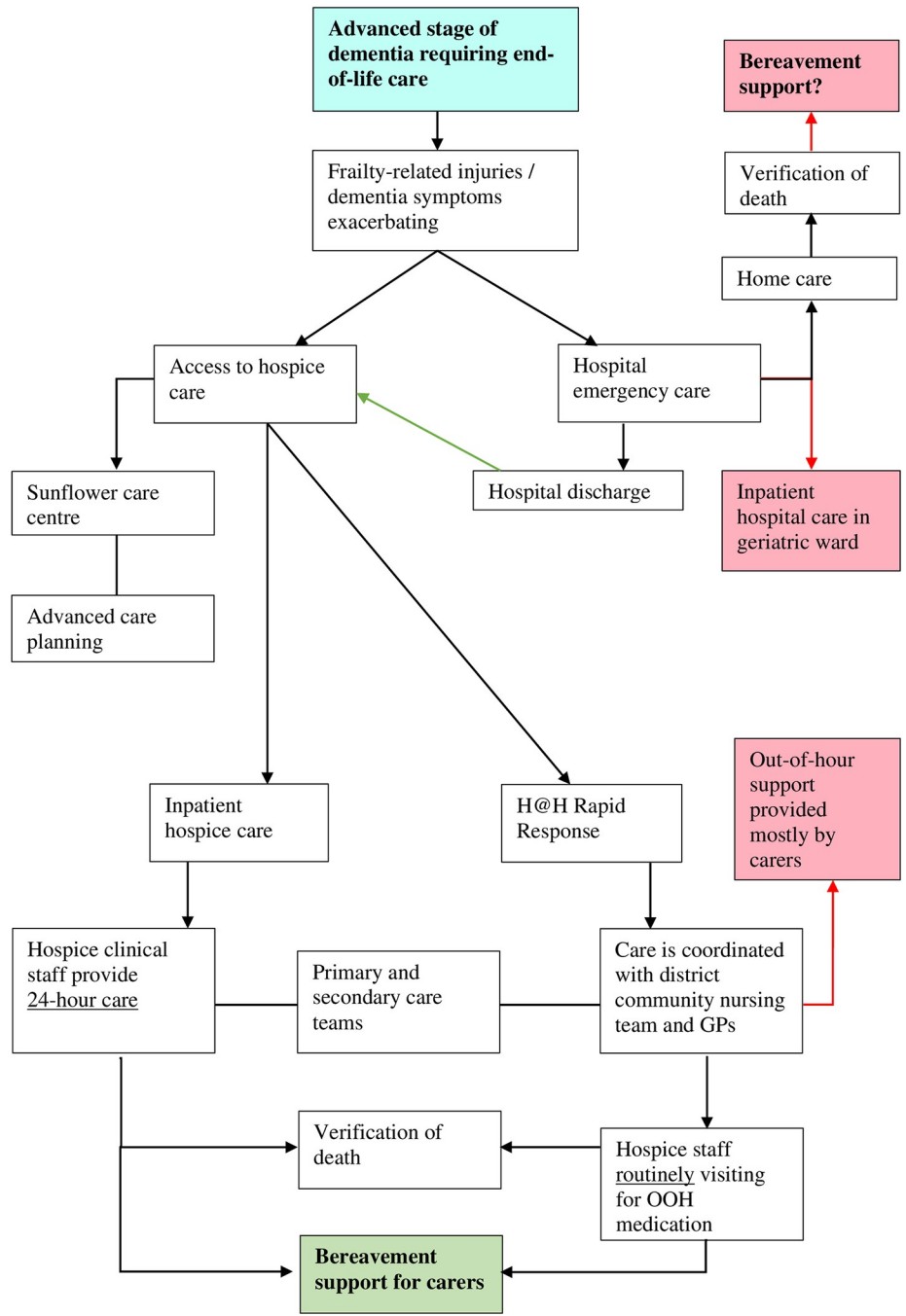

**Fig 1. East Cheshire end-of-life and palliative dementia care pathway.**

homogeneous group of participants. We based our sample size on conceptual density (i.e., the collection of data until a sufficient depth of understanding is reached) [18], which was agreed upon by all study authors. We delivered semi-structured interviews with participants and reached sufficient depth of understanding (data saturation) with 12 interviews. Out of 12 participants, seven clinical hospice staff and five family carers, all were female and white British from one service in the Northwest region. Two staff were medically trained in general practice,

three were nurses with one being a palliative care specialist, two were health care assistants. The interviews lasted on average 38.30 minutes.

The interview with carers focussed on their experience of accessing end-of-life care in dementia and providing care during later stages of the condition (e.g. coping mechanisms, quality of care from the service). An example of the questions we asked carers was: is there any aspect of the advanced care planning that you think should have been explained differently or could be improved?

The interviews with clinical personnel focussed on barriers and facilitators they encountered when managing people living with dementia at the end of life. An example of the questions we asked clinical personnel was: I understand you provide both inward and homecare, from your knowledge and understanding of hospice care, do you think that inward care is something different from homecare? And in what ways?

## Data analysis

Thematic analysis [19] was used to analyse all interview transcripts, by inductively identifying and reporting themes or patterns within the data set. The analysis was facilitated by the use of Nvivo software 12 in that it enabled to organise and highlight emerging patterns (codes). Upon identification of initial codes, the researchers AB and CDL defined themes and sub-themes in relation to the objective of the study.

To increase accuracy in reporting, a codebook was created by AB and used by two researchers (AB and CDL) to code the interview transcripts by having a similar instrument for coding. Three versions of the codebook were necessary to refine the process of coding.

## Results

Three main themes and associated subthemes emerged from the thematic analysis (Table 1).

### 1. Pre-access to service

This theme refers to the times clinical hospice personnel and carers commented on their experience of pre-access to hospice care. The experience was reported in terms of having advanced care planning in place at the time of accessing the service, barriers to access and how hospital care was often a first option for people with dementia.

*1a. Advanced care planning.* Clinical personnel and family carers described the ways advanced care planning was made before access to hospice care, and strategies that were in place to understand the wishes of the person with dementia in critical times.

One healthcare assistant explained that having proper discussion around advanced care planning may be difficult at times to achieve given the very last-minute admission made to the hospice:

*'. . .If somebody's not in the hospice, that's done in the community by Macmillan or district nurses. Then we do have ongoing talks when people come in, because not everybody has made those plans. People can come in at short notice, so not everybody has got plans in place. We do have discussions with people about their wishes and what they want to do. . ..'* (Staff 7 –Clinical specialty in health care).

The sunflower centre is a day centre for patients and their families to spend some quality time together. It would act as bridge from early dementia diagnosis to later stages of the condition when hospice care may be needed:

**Table 1. Theme categorisation.**

| Theme coding assigned to each theme (N) | Subtheme | Participant's account |
|---|---|---|
| **1. Pre-access to service (n = 18)** | *1a. Advanced care planning* | 'Yes, I mean sometimes they're, the patients that we admit have been known to the hospice before, through our day care unit and if they have been known to the hospice through day-care then often our advanced nurse practitioners who generally see our patients on day care will have had those, or some of those advanced care planning discussions with the patients before they end up coming in as an inpatient.' (Staff 1 –Clinical specialty in general practice). |
| | *1b. Barriers to access* | '. . .She got sepsis. And then she went to the A&E. . .Initially it was good, and then it was terrible. She was not getting proper analgesia, mouth care, turns, it was just terrible. And I begged them to get her to the hospice several times. The consultant said she "Wasn't in the dying phase." This was four days before she died.' (Carer 5 –Mother had dementia). |
| **2. Roles and responsibility within hospice care (n = 22)** | *2a. Working at the hospice* | '. . . So I work in the hospice on the inpatient unit. So we're a 15 bedded unit and my routine would be we have a get together with the other multi-disciplinary team in the mornings. So the nurses, the physio, the OT, the social worker, the chaplain, whoever's around really and talk about each of the patients on the ward and get an update.' (Staff 2 –Clinical specialty in general practice). |
| | *2b. Type of dementia support in end-of-life and palliative care* | 'On the ward all of us provide care for people with dementia. In the Sunflower Centre, yes we do. So basically, while the carers go off and do the eight-week course. . .I am in a separate group doing activities with the people that they care for. Obviously, that includes taking them to the toilet, making sure they have drinks and some food. Just sitting with them and reassuring them, doing activities.' (Staff 6 –Clinical specialty in health care). |
| | *2c. Being a carer in end-of-life dementia care* | '. . .so mum's kind of verbal communication was quite limited by that stage. I think the first couple of days she could still speak a little, but it was a little, but she, I always knew if she was particularly anxious because she would get hold of my hand and squeeze it and shake it, so I'd come to rely on that really, I knew that if she was bothered she would do that.' (Carer 3 –Mother had dementia). |
| | *2d. Opting for H@H Rapid Response versus inpatient care* | '. . .Yes. I think that's another thing which differs to hospital. I mean not everybody, but we've got some patients who really do cover the walls with all the stuff that is personal to them. You know, their own blankets, their own clothes obviously, photos and pictures. . .' (Staff 2 –Clinical specialty in general practice). |
| **3. Ease and difficulty of last period of end-of-life care (n = 23)** | *3a. Discharge pathway* | '. . .You know, they might have gone into hospital for something and their sort of general function is more poor. So whatever that person was doing beforehand, wherever they were, might not be suitable now. So the nurses would like be doing a sort of 48 hour care plan, looking what their nurses needs would be. And then so we've got a social worker on board as well so the social worker would be looking at what care might be available to the patient. And then the OT would do an assessment for either an environmental visit or a home visit and that would involve the relatives as well. And then if it's particularly kind of complicated we have a discharge planning meeting.' (Staff 2 –Clinical specialty in general practice). |
| | *3b. Key features providing PCC* | . . .'I had to think about it being infection-control friendly as well, things that we could wipe during COVID. We sourced books, jigsaws, twiddle muffs, handbags, purses, some money, different stuff like that that they could have, that they are able to have in their room if they want it.' (Staff 7 –Clinical specialty in health care). |
| | *3c. Bereavement support for carers* | '. . .dependant on the need at the time we can provide informal pastoral and psychological support. We do have access to a variety of talking therapies and pre-bereavement support as part of hospice services who we can refer relatives to should we feel they need and they would like to be referred on for.' (Staff 4—clinical specialty in nursing). |

'. . .I think a lot of that is down to the work that Sunflower or day hospice is doing. Because we're seeing patients in the earlier stages of dementia, because they're accessing services earlier on. Hopefully it means that when they come to end of life or have symptom control problems that they're remembering that we're here.' (Staff 5 –Clinical specialty in nursing).

In east Cheshire region, there is a sharing of information about patients' wishes, between primary and secondary services that occurs through EMIS records, this would act as joined up care to ensure patients' wishes are taken into consideration as part of end-of-life care:

*'. . .if a patient in the Sunflower Centre makes a decision regarding, for example resuscitation then I would contact their GP. . .We also have an EPaCCS system which is a template on our computer the EMIS record. . .which is a shared record through the community and the hospice. Which you can put on advanced their planning wishes.'* (Staff 5 Clinical specialty in nursing).

Responsibilities to make key decisions are based on the type of existing needs for the person, as different needs may require a slightly different decision pathway to be in place:

*'So, it depends on the decision, so if it was a decision to not want to be resuscitated that would need to go through the GP first. If it's a discussion about preferred place of care, for example or preferred place of death. Then I would log that on EPaCCS first because it doesn't need to go through the GP.'* (Staff 5 –Clinical specialty in nursing).

Carers reported experiences of advanced care planning. One carer explored this with their grandmother with dementia early on in disease trajectory and this ensured that key decisions were made in line with the patient's preferences:

*'She had done her own advanced care planning with my dad, who was the executor of her will, so we knew what she wanted in terms of care, that was prior to her dementia being so severe that she couldn't tell us. But we knew that she wanted to stay at home for as long as possible, she didn't really want to be in a nursing home and she had agreed to a do not resuscitate form when the GP had spoken to her about that.'* (Carer 1 –Grandmother had dementia).

She continued, however, that because the symptoms did not improve at the hospital, she felt the need to have the grandmother referred to specialty end-of-life care to help her have the type of care that was needed at the time:

*'. . .So she was in hospital for a week having active treatment for a chest infection and after a week I spoke to the clinician on the ward and the medics and said, "This is ridiculous. She's not getting any better, she is not communicative, she is actively dying. I want her at the hospice." We knew about the hospice care and we asked for a referral over to the hospice.'* (Carer 1 –Grandmother had dementia).

Another carer commented on how dementia impacted her mother's ability to engage in autonomous decision about care planning. She, as a carer, had to make informed decision based on the advice that was provided to her by the family doctor:

*'. . .mum. . .was ill quite quickly and the doctor was very good. Did really kind of quite strongly advise me not to take mum home, because I would have been on my own it would, we'd have not had social care support, so yes. . .'* (Carer 3 –Mother had dementia).

*1b. Barriers to access.* Some carers reported barriers to hospice access which were explained in terms of lack of proper funding to support beds availability within the hospice, and hospital as first care option available in emergency situations in end-of-life care.

One carer reported how difficult it was to have a referral to proper hospice care: *'. . .there is not much really that could be done for that, to give more funding to the hospice so that there are more beds available really.'* (Carer 3 –Mother had dementia).

She continued that a solution to the problem could partially depend on little government funding currently devolved for end-of-life and palliative care services:

'. . .it wasn't anybody's failings, it was just the lack of beds really, so you know, apart from like I say, apart from the government deciding they will give more money or something like that, I don't think anything really would have made much difference.' (Carer 3 –Mother had dementia).

Another carer described how with frailty in dementia it is often the case that the person is first admitted to hospital for a fall-related injury. This coupled with little knowledge of hospice dementia services, delayed the decision to have the mother-in-law referred to hospice care:

'. . .During the time after she was diagnosed with dementia, she had a fall and was taken into hospital, but then became quite violent so she was actually sectioned. And then for the last-just under a year of her life, she had to go into a care facility nursing home because she wasn't safe to be on her own. . .and I never realised that the hospice actually took patients that had dementia.' (Carer 4 –Mother-in-law with dementia).

Barriers in access to hospice care were reported by a carer who had her mother hospitalised for an infection and they felt their mothers end-of-life care needs were not being met. Only after the carer made a complaint, her mother was transferred to hospice:

'And I said, "Why are you giving her a short-acting injection? Has she not had her long-term morphine?" And they said, "No." I said, "Why?" And she said, "Oh, because they're doing an audit." And I just lost it then. I said "Get my Mum to the hospice now."' (Carer 5 –Mother had dementia).

**2. Roles and responsibility within hospice care.**   This subtheme refers to the roles clinical personnel and carers assume within the hospice environment. There are times when the daily demands posed by end-of-life care require some adjustment to the clinical role, with some staff experiencing an extension of their usual clinical input, be it psychological support or act as champion for carers' wellbeing. A transition in caring role seemed to be experienced by carers when hospice support was provided.

*2a. Working at the hospice.* A clinical staff member with background in general practice described her working routine, which seemed to be rich in tasks around patient's admission, assessment, providing out-of-hour care, but also training new workforce:

. . .So I work two days one week, three days the other week in the hospice. Generally the day starts with a handover for the virtual ward round and then the rest of the day is made up by seeing the patients on the ward, admitting new patients, supervising trainees, occasionally going to day care and I do on call, covering the in-patient unit overnight every other Monday, some weekends and sometimes have to obviously answer advice line calls that come into the hospice either in hours or out of hours. (Staff 1 –Clinical specialty in general practice).

One nurse explained having quite a broad spectrum of responsibilities, with some more direct contact with patients compared to medics, by providing, for example, a first line of general psychological support:

*. . .I work as part of a small team of doctors undertaking daily medical admissions, reviews and discharges of patients. That includes review of physical symptoms and making adjustments to medicines, regimes and providing psychological support, education and basic counselling.* (Staff 4 –Clinical specialty in nursing).

To ensure that effective care is provided, a clinician explained that there is usually a limit to the number of daily admissions made to the hospice, as to ensure that attention and right amount of clinical input is effectively provided to newly admitted patients:

*. . .And then I would sort of prioritise seeing whoever I need to see first and also we talk about whoever might need to be coming in for the day, admitted onto the ward, and we'd have a maximum of two admissions during a day.* (Staff 2 –Clinical specialty in general practice).

*2b. Type of dementia support in end-of-life and palliative care.* Roles and responsibilities of clinical staff were not limited to inpatient care at the hospice, as some nurses were also involved in providing medical cover for the Sunflower centre, which is a day centre for people from the community:

*'I also provide medical cover for the Sunflower Centre which is our day hospice so we also have patients with dementia over there as well.'* (Staff 5—Clinical specialty in nursing).

The sunflower centre is programmed to reach out for carers as well as the person with dementia, by providing a safe space for us-time for them:

*'Yes, they have different programmes. They have a dementia carers programme, which I think is an eight-week programme where carers and the person with dementia attend together. . . We also have a carers group, which is like a carers' support group. We have a singing, like a choir, dementia choir.'* (Staff 5—Clinical specialty in nursing).

A health care assistant explained that being involved in the hospice daily activities means to volunteer and support colleagues in the running of dementia carers groups, for example: '*on Wednesdays when that dementia carers group runs for the eight weeks, I always go and help out on that.*' (Staff 6 –Clinical specialty in health care).

This type of support was often challenged by the very unpredictability of dementia as a condition, whose severity could change over time:

*. . .And, of course, you have to be on the ball because you can have a group of 15 people and when you meet them for the first time you have no idea. You might have on your handover sheet, -This person only got diagnosed recently or this other person got diagnosed maybe five years ago- but until you meet them, you don't actually know how they are going to be and how they are going to mix with other people. You have got to very quickly. . .'* (Staff 6 –Clinical specialty in health care).

At advanced stages, dementia may limit the quality of communication and make it difficult to clearly understand patient's needs and preferences. Checking patient's history and talking to family may help find a suitable care option to comfort care. This was shared by a healthcare assistant who felt that the quick changes in dementia symptoms and severity would require a day-to-day change in care to ensure PCC is promoted throughout:

*'Quite often people have got families that can give us background into the patient. You just have to treat them on an individual basis. It's not even on a daily basis, it's an hour by hour because their behaviour can change so rapidly, can't it?'* (Staff 7 –Clinical specialty in health care).

*2c. Being a carer in end-of-life dementia care.* Carers reported instances when having to make prompt care decisions in emergency situations proved emotionally exhausting:

*'And then because I couldn't give her the care that I needed, I felt guilty about taking her into hospital. And she said, "You said- You promised you wouldn't," and I'm like, oh, God. You know, it's terrible.'* (Carer 5 –Mother had dementia).

One carer disclosed that the experience pre-hospice care was just: *'a terrible time, a lot of it I felt as if I didn't take it all in.'* (Carer 2 –Mother had the dementia)
Being a carer meant to be firm in the choice to take the loved one at the hospice:

*'. . .the ambulance crew said to me, "Are you sure you want your nan to go to the hospice?" Because she was so unwell they weren't sure whether she would make the short journey in the ambulance from the hospital to the hospice. And I said, "Just get her there." I got in the back of the ambulance, and I said, "Just take her and we'll deal with whatever happens on the way." So I prayed very hard for the 10 minutes that it took us to get here, but the good care that she had, she lasted another week. . .'* (Carer 1—Grandmother had dementia).

Exploring care expectation was reported by a staff member as key to ensuring that proper care is provided:

*'Very much so and we do try to do that with everybody on admission because in the past we've found that actually some patients and carers expectations have been way off what we can actually deliver and what our remit is. So yes we do try to sort of you know set the scene on admission.'* (Staff 4—Clinical specialty in nursing).

A carer, however, felt that being asked about expectations of care would not be considered a priority in end-of-life care, as the goal setting would be around comfort care rather than symptoms remission:

*'. . .I was just grateful for her to be somewhere where they would looking after her and putting her at the centre. So I don't think, I am kind of, from my experience, you know, I would say yes that would have been a good thing to do, but in reality I don't think that would have made any difference.'* (Carer 3—Mother had dementia).

*2d. Opting for hospice at home (H@H) Rapid Response versus inpatient care.* H@H Rapid Response service was explained by a hospice member of staff in terms of ad-hoc type of care:

*'You need us, pick up the phone.* (Staff 3 –Clinical specialty in nursing).

The nurse then explained that hospice care is quite different from the type of support patients would normally receive from district community nursing teams, as the latter would regularly visit them on a daily basis, whereas hospice clinicians would be called into action when needed:

*'. . .The district nurse would only come if there was an immediate nursing need, say like they needed pain relief by injection. The hospice at home team, we don't actually provide regular calls. . .We don't say to our customers, look, we will come every night at, say, . . .if we do that, that means that our ability to respond to urgent requests for medication or deal with some-body who's fallen just dissipates.* (Staff 3 –Clinical specialty in nursing).

One staff described the situation when H@H Rapid Response care is preferred:

*'You know when you've got somebody with dementia who's perhaps fairly active who needs activity to keep them occupied rather than wandering aimlessly and getting themselves in to trouble, you know the idea perhaps that a purely clinical environment might not be appropri-ate for them. So as I said before a home environment that they're familiar with, with their own surroundings and belongings might be more amenable, more appealing, actually the bet-ter place to provide care for somebody in some instances.'* (Staff 4—Clinical specialty in nursing).

Whereas there may be advantages of having either option of care, inpatient care seems to offer better access to medications during out-of-hour care:

*'. . .Medication wise for us it would be different. Thinking of things at the root of administra-tion and medications, what would suit them. What you might not give to a patient earlier on in their disease because you'd be worried about side effects actually you've kind of at the end of life it's more important to keep them symptom controlled. So you might use a medication, you know, perhaps let's say Haloperidol for nausea and vomiting or something like that but you might not use when they're earlier in their disease [in the early phase of the disease].'* (Staff 2 –Clinical specialty in general practice).

One carer explained that although H@H Rapid Response service is preferred to inpatient care, as it would maintain the person within the home environment, the access to 24-hour medication is not available:

*'. . .I think the home-based care . . .In my personal opinion, is what we should be aiming for, because people that want to die at home want to have the quality that you provide in the hos-pice- They want that at home. But the people aren't there. It's not responsive enough. I had to go and get all the medications from the GP, we had to get the blue book signed, I had to call 111 to get my Mum's medication elevated.'* (Carer 5 –Mother had dementia).

**3. Ease and difficulty of last period of end-of-life care.** This theme refers to a specific discharge pathway in place for end-of-life hospice care, but also the overall experience of clini-cal personnel and family carers of getting close to the end of care.

*3a. Discharge pathway.* Hospice staff and bereaved carers reported their experience of dis-charge from hospice. The discharge process represented a worry for carers who would feel unprepared to provide home-based palliative care as alternative option, whilst for staff, this was accepted in the case of symptoms improvement or in the presence of very long stays which would not require close-to-death inpatient support.

*'. . .if someone was being discharged then on the discharge letter that would go to the GP, and then the sort of community nurse would have access to that if they wanted. And we've got the*

*dementia nurses here so if they were involved as well they would be able to read it. And on that it's got a section on preferred place of care, preferred place of death, resuscitation, choices. So it would be kind of documented.'* (Staff 2 –Clinical specialty in general practice).

One of the clinicians in the hospice explained that discharge would also occur if the person with dementia was first admitted to hospice care for their symptoms management which then seemed to improve through their professional support:

*'. . .I think generally in that case they are with us for end of life. I think some patients would come to us with maybe you know, co-morbidities, dementia and a cancer for example and if we, through being with us we achieve or they achieve better symptom control and you know, there's an opportunity for them to go home, because they're not imminently end of life or go, you know, be discharged really, then obviously we would aim to do that if it, if that's in, you know, in keeping with their wishes, either whether that is to home or to a nursing home and then obviously when planning their discharge we would have to take the dementia diagnosis and the symptoms and the issues that go with that into consideration when planning their discharge.'* (Staff 1 –Clinical specialty in general practice).

There are instances when the family carers were so pleased to receive hospice care for their loved ones, that they would then be worried about discharge:

*'. . .we have families offer to pay to keep their loved ones here for longer, that happens all the time, but it's not something that we offer or it's not something we accept.'* (Staff 5—Clinical specialty in nursing).

Only one carer reported the experience of discharge (Carer 2 –Mother had dementia). Although discharge was only in discussion and never occurred, the risk of not knowing what would be next for them, increased carer's worries for future care.

*3b. Key features providing PCC.* Staff and carers reported some features of the care offered at the hospice as key to promoting person centred care in the person with dementia:

*'. . .considering a patient as an individual, providing holistic care, being aware and having the wish to support the family, and to, yes, being aware and, you know, doing our best to support a patient, manage symptoms appropriate, acknowledging the diagnosis. . .'* (Staff 1 –Clinical specialty in general practice).

Strategies to support good quality care, entailed staff repeating the breathing alongside them:

*'I think yes it is, it depends how bad their dementia is but I suppose you could if you breathe with them perhaps they might mimic you. So if they're breathing quite fast but if you're sat in front of them and they can sort of see your breathing then that might sort of slow them down. But I would just manage otherwise if I was using medication I'd just use it the same that I would do with someone else.'* (Staff 2 –Clinical specialty in general practice).

Carers reflected on how the person-centred care provided in the hospice created a sense of peace in knowing that the loved one was receiving proper compassionate care:

*'. . .it was that completely patient centred care, the kindness, the respect, the constantly thinking about her as a real individual. And getting to know her. . . just made me think okay just*

*you know, you've got to keep calm here. . ., so yes, so really good at keeping that calm.'* (Carer 3 –Mother had dementia).

One carer, acknowledged the quality of care provided by the hospice staff, however reported:

*'. . .it's difficult to answer that, really, because she wasn't really [audio distorted 00:15:51] she was sort of barely conscious most of the time. As I say, they were very attentive, they came in very regularly to check in her.'* (Carer 4 –Mother-in-law with dementia).

Another carer reported being particularly impressed by the prompt intervention the clinical staff provided to patients. This was described as compassionate and attentive towards the needs of the person:

*'. . .I think just the general love, care and attention that she didn't get at the hospital. She wore the same nightie at the hospital for the whole week that she was there and at the hospice she didn't, you know, they gave her mouth a clean every couple of hours.'* (Carer 1 –Grandmother had dementia).

*3c. Bereavement support for carers.* Family carers and staff reported on the type of bereavement support offered from the hospice after the person with dementia had passed away:

*'. . .We get family members that have lost loved ones that still pop in for a chat or a cup of tea or bring cakes. We keep in touch. We do a time to remember service as well, so three months after somebody had passed away, they are invited to like a service in memory for the people who have died within that period.'* (Staff 7 –Clinical specialty in health care).

One carer reported that bereavement support was effective for them and briefly explained the chaplaincy service at the hospice:

*'. . .they would just come in and check you were okay and chat with us about anything and everything, really. She would often go and bring us back a cup of tea and just befriending people, really. And if they wanted to talk about religion, then they would talk about it. But if not, they would just come in on like, a friendly basis.'* (Carer 4 –Mother-in-law with dementia).

Despite the fact that the nursing team was very attentive, a carer reported that more support should be provided for her:

*'I think the nurses were very good, they would kind of talk to you but obviously they're having to talk to you around delivering that care. So that is quite difficult. Whereas I think if I had had somebody that could have just sat down and talked to me for half an hour or so, that would have been really helpful.'* (Carer 3 –Mother had dementia).

Although hospice support for carers extend well after the person is passed away, one carer believed that hospice support would be available only during the time of active care for the patient: '*It was there. And they did say, you know, that you could, but I think you feel like because they're no longer here*' (Carer 5 –Mother had dementia).

## Discussion

This was a qualitative study exploring the experience of bereaved carers and hospice clinical personnel of hospice dementia care. We employed thematic analysis for theme categorisation

and data interpretation. From the available data we found that the service used joined up care with primary and secondary community services to ensure a smooth running of care for the patient and their families, but also that information about patient's preference for end-of-life care was effectively shared across health care settings. The service seemed to plan care in a flexible way to promote PCC even in situations where verbal communication was impaired by the advanced stage of the condition and alternative ways to understand patient's needs were in place.

In concert with existing evidence [20, 21] and as reported by hospice clinicians in our study, the uncertainty of dementia progression to reach an end-of-life stage may require more specialist type of end-of-life care, compared to home-based support. This was echoed in bereaved carers who shared that although their preferred setting would be home, there were times where having support from the R&R rapid response team could not cover out-of-hour needs and the carers would become the main referee for the patients' care. The high availability of health care professionals found in inpatient hospice care was found to be beneficial for the patient and their families, as more consistent 24/7 specialist support would be provided.

A key feature in the model of care hospice clinicians reported using was to ensure active collaboration with other professionals across services. This would promote joined-up care between primary and secondary healthcare services through information sharing, especially in relation to important patient's information such as preferences for treatment plans. In line with evidence [22, 23], in our findings, neither clinicians nor carers reported preferring invasive/intensive care as an attempt to prolong the life of the patient, they instead discussed about the importance of comfort and compassionate care through holistic care. This seems to be in contrast with findings from a previous study on home care unit professionals' attitudes towards palliative care, with the authors reporting that health care professionals opted for more intensive care and this was also a preferred care option over palliation for caregivers [21]. The study authors, however, report the difficult care environment that home care unit professional operate in, when managing cases, as they face staff shortage and operate with little resources on a daily basis (e.g., short visits to each patient's home) to effectively manage end-of-life patients. Also the study was conducted in Israel with a different healthcare system from the UK, leading to potentially different set of barriers/facilitators impacting on the experience of care.

This study has strength in the methodology used for data collection and analysis. Multiple raters coded the interview transcriptions and theme categorisation was derived inductively through multiple reiteration of schema coding. A future study will be needed to extend this exploration to hospices elsewhere in the UK. This is particularly important given the heterogeneity of service provision for end-of-life care which varies across geographical locations. Furthermore, it is the case that a minority of patients, and an even smaller minority of those with dementia, die in a hospice or receive hospice at home care. Studies such as this need to be repeated in settings such as acute hospitals, psychiatric hospitals, nursing homes and non-hospice community services which support people to die in their own homes. The study has some limitations and therefore study findings need to be taken with caution. For example, we managed to recruit only white British female participants and a more diverse sample (e.g., based on ethnicity, gender) would have maximised our exploration of differences in needs and preferences of care. In addition, we recruited a small sample as only one hospice acted as recruitment site, however, it was not our intention to extend our recruitment to multiple sites as the aim of this empirical work was to engage in a qualitative scoping exercise around the experience of dementia hospice care.

## Conclusion

No unique care pathway exists for end-of-life dementia care and people may be disoriented by the number of options to choose from. Here we identify positive experiences and challenges

for staff and carers of providing hospice support to those with dementia at the end of life. Further work should focus on the role of the hospice in end-of-life care for dementia as part of a larger set of services and how benefits identified here can be maximised whilst minimising challenges.

## Author Contributions

**Conceptualization:** A. Bosco, M. Dunlop, A. Booth, C. Todd, A. Burns.

**Data curation:** A. Bosco, C. Todd.

**Formal analysis:** A. Bosco.

**Funding acquisition:** A. Bosco.

**Investigation:** A. Bosco.

**Methodology:** A. Bosco, C. Di Lorito, D. Alexander, S. Jones, A. Burns.

**Project administration:** A. Bosco.

**Resources:** A. Bosco.

**Software:** A. Bosco.

**Supervision:** A. Bosco.

**Validation:** A. Bosco, B. R. Underwood.

**Visualization:** A. Bosco.

**Writing – original draft:** A. Bosco.

**Writing – review & editing:** A. Bosco, C. Di Lorito, M. Dunlop, A. Booth, D. Alexander, S. Jones, B. R. Underwood, C. Todd, A. Burns.

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
