## [Decision Letter · Decision Letter 0]

2 Jul 2023

PONE-D-23-14587The East Cheshire Hospice (Teach) Dementia Programme: Increasing awareness about end-of-life care.PLOS ONE

Dear Dr. Bosco,

Thank you for submitting your manuscript to PLOS ONE. After careful consideration, we feel that it has merit but does not fully meet PLOS ONE’s publication criteria as it currently stands. Therefore, we invite you to submit a revised version of the manuscript that addresses the points raised during the review process.

We look forward to receiving your revised manuscript.

Kind regards,

Ayi Vandi Kwaghe, D.V.M., M.V.Sc., P.G.D.E. Ph.D., MPH

Academic Editor

PLOS ONE

“Yes. Data collection occurred in the premises (hospice) where the funder is located.”

Additional Editor Comments:

-Please, remember to put the continuous line numbering on the entire manuscript and the manuscript should be well spaced. This will enable ease during review and editing as it allows the reviewers and Editors to easily point out errors and corrections needed in the manuscript.

Abstract

Results

-This part of the abstract should be under methods since it is reflecting the number and type of participants that were recruited and interviewed “A total of 12 participants were interviewed from one service in the Northwest region in the UK. All were female and white British.”

-Clearly state the generated themes and the major challenges encountered in your study and delete this statement “Themes identified clustered around the experience of pre-access to hospice care, role and responsibility of staff and carers within hospice, and the challenges of delivering care during the last days of life for the person with dementia.”

-Conclusion

State clearly what need to done to improve the services

Background

-I released that the title background/introduction is missing in the manuscript.

-The research question should not be a heading after the background but part of the background information. “Primary Research Question: What is the experience of end-of-life hospice care for bereaved carers of people with dementia?”

-Please, ensure that the objective or aim of the study is the last sentence in the last paragraph of your background/introduction

Methods

-Shouldn’t the participants be given full information on the study before signing the consent form? Why giving them after the interview? As stated below in your manuscript:

“All participants who agreed to take part in the qualitative interviews were required to sign a consent form and at the end of the interviews they were provided with a debriefing form summarising the aims of the study and how to address any issues (if any) that could occur during or after the study. Bereaved carers were only approached if a minimum of”

-Take a paragraph each to express the eligibility criteria for carers and hospice clinical personnel instead of using bullet points.

-The statement for data storage should be under data availability and not part of the methods.

-What were the methods you used to eliminate or minimize bias? Inductive coding and or reflexivity?....

-What type of coding method did you use? From the study I realized that you used inductive method of coding. State it clearly in your methods.

-Nothing was mentioned with regards to “data saturation point”. Did you at any point of the interview achieve data saturation? Please state whether or not this was achieved as part of the methods. If data saturation was not achieved during the interview, kindly state it as part of the limitation of your study.

Results

-The statement below should be part of the methods involved in recruitment of participants and should not be part of the results. Please, insert this section under methods in the appropriate location:

“Semi-structured interviews were held with 12 participants (seven clinical hospice staff and five family carers, all were female and white British) form one service in the Northwest region. Two staff were medically trained in general practice, three were nurses with one being a palliative care specialist, two were health care assistants. The interviews lasted on average 38.30 minutes.”

-Please, do include a table that will clearly indicate the THEMES, SUBTHEMES, CODES GENERATED and REPEATITION OF CODES by participants for more clarity on the data analysis and understanding of the data at a glance. This table should be in the result section and modified as indicated (Table 1). Participant account may not be necessary in the Table, it has already been stated under the various themes and subthemes in the result section.

-Covid should be written as COVID all through the manuscript please.

-Do you mean in the early phase of the disease (dementia)? “when they're earlier in their disease”, you may need to rephrase the statement for clarity.

Please paraphrase the statement for clarity - “So I think about half patients come in for symptom control and then probably the other half of patients come in for end-of-life care.” Do you mean half of the incoming patients?

-I do understand that the translations and transcriptions were done verbatim but some of the transcriptions need to be properly translated for clarity and at the same time making sure that you do not alter the meaning of what the participant said. For example the statement from one of the participants below located in pages 12 and 13 of the manuscript:

“…I think generally in that case they are with us for end of life. I think some patients would come to us with maybe you know, co-morbidities, dementia and a cancer for example and if we, through being with us we achieve or they achieve better symptom control and you know, there’s an opportunity for them to go home, because they’re not imminently end of life 13 or go, you know, be discharged really, then obviously we would aim to do that if it, if that’s in, you know, in keeping with their wishes, either whether that is to home or to a nursing home and then obviously when planning their discharge we would have to take the dementia diagnosis and the symptoms and the issues that go with that into consideration when planning their discharge.’ (Staff 1 – clinical specialty in general practice).”

Discussion

-Please, delete all the statements highlighted below because the statement is about the methods and partly results. Just discuss your findings straight ahead. The statements have been dealt with earlier on in the manuscript and there is no need for repetitions. Do not put Figures in discussion please.

“This was a qualitative study exploring the experience of bereaved carers and hospice clinical personnel of hospice dementia care. We employed thematic analysis for theme categorization and data interpretation. From the available data we found themes around the experience of pre-access to hospice care, during care delivery and around the bereavement support available to carers soon after the death of the person with dementia. The data allowed us to have a better picture of hospice care pathway that is currently being offered to people with dementia in one of the settings in the Northwest region in England (Figure 1)”

-Do not put headings in your discussion, simply use a paragraph to properly explain your points in the discussion.

-Please, use the last paragraph and discuss the limitations of your study. The strength and future prospects this study should be discussed in a separate paragraph.

-I realized that there was no citation in the discussion. Despite being an empirical study, I believe there may be some articles about dementia patients and some findings that may be related to your study which can be cited as part of your discussion.

Reviewers' comments:

Reviewer's Responses to Questions

**Comments to the Author**

1. Is the manuscript technically sound, and do the data support the conclusions?

Reviewer #1: Yes

Reviewer #2: Partly

2. Has the statistical analysis been performed appropriately and rigorously? 

Reviewer #1: Yes

Reviewer #2: N/A

3. Have the authors made all data underlying the findings in their manuscript fully available?

Reviewer #1: Yes

Reviewer #2: Yes

4. Is the manuscript presented in an intelligible fashion and written in standard English?

Reviewer #1: Yes

Reviewer #2: Yes

5. Review Comments to the Author

Reviewer #1: This is a really important and interesting paper. I have made some comments throughout in a bid to strengthen the analysis and message of the paper and hope the authors find them useful. For quickness the comments are blunt.

Title

Based on the title I was surprised when the abstract revealed it was a qualitative study. Could the title be rephrased to include study design and more accurately represent the paper (e.g., Experiences of increasing awareness about end of life care for those experiencing dementia: Qualitative study of carers and hospice staff). Not sure the detail about East Cheshire is needed and isn’t picked up in the abstract.

Abstract

The conclusion should be stronger (more work is always needed) – what is this study’s unique contribution?

Introduction

Nathan Davies at UCL has done quite a bit of work around decision making at the end of life for those with dementia. Although you touch on some of his work, I would bring more of his recent work in here.

It would be useful to explain the differences in care between care homes and hospices for people with dementia (and therefore this justifies why hospices should be explored specifically).

Aim is different to the one in the abstract (brings in role of the hospice and use of PCC). I would suggest removing the primary research question and have one clear aim in the abstract and end of introduction. The PRQ also only concerns carers not staff.

Methods:

Would be useful to have some subheadings in the methods (e.g., participants, inclusion/exclusion criteria, data collection and data analysis).

Add ‘(PPEI)’ in page 4 after spelling it out.

Add Participant Information Sheet on before PIS at the starts of page 5.

Can the hospice be described in more detail to set the context (e.g., how large, is it in a deprived or affluent area, is it dementia only).

Eligibility – why did carers have to be over 50? The reason should be included.

You’ve stated a minimum of 4 weeks had to elapsed (should be added to eligibility criteria), but was there a maximum amount of time (e.g., within the past two years)?

The interviews with hospice staff focused on barriers and facilitators to managing people living with dementia – this is the first time this aim or focus is mentioned in the paper. It needs to come earlier and thread through. Again the interviews with the carers focused on accessing. The title states ‘increasing awareness’ and the aim is around role of the hospice and use of PCC. Perhaps there needs to be one broad aim (e.g., experiences of end-of-life care in hospices) and then several objectives underneath (e.g., role of hospice, management, barriers and facilitators).

Add a reference for thematic analysis as there’s been many updates and different approaches.

I would scrap the inter-reliability check and have more confidence in your qualitative analysis. I do not think qualitative data or the process behind it needs to be quantified unless there are concerns. It may be that carers and hospice staff have different views of this complex time and that would be perfectly okay.

Results

A table or summary of the characteristics of participants would be useful of anything other than gender and ethnicity (e.g., age, length of time since bereavement, years of experience, job role).

Unsure why the East Cheshire end-of-life pathway figure is in the results and discussed in the discussion – from what I can see this should be introduction or methods. This comes out of the blue.

Again I would lose the quant reliability check and explain how qualitatively you ensured this data was robust (e.g., in the methods include some reflexivity - have multiple professions/backgrounds looked over the themes?). Include this information about the authors.

Somewhere you need to address the sample size between your aim and your final number. If you had challenges recruiting then this is important to highlight and can be helpful. Also is this hospice area predominately White British or did different ethnicities decline to take part (discussing end-of-life can vary across cultures)?

Results second line – should be from not form.

The results are quite list like and could be more synthesised. Rather than relying too much on the quotes the explanation can come through from the authors and then the quotes highlight and add context/detail to the text. Rather than a list of quotes. Sometimes this is done well, but take a look through to see areas where more elaboration/synthesis can be given (e.g., Page 9 ‘2a a clinical staff member with background in GP describes her routine.’ Here I would say what the take home message is for the reader then present the quote).

Halfway down page 7, carers report different experiences and then two examples are given, what were the other three experiences? Did most have advanced care planning?

The first part of the results (1a) seems to highlight a system that is working well with joined up care (e.g., sharing of advanced care plans), working with the community. I would talk about that here.

1b is an interesting sub-theme with good interpretations.

A table or figure might be a good way to present the data at the start of the results. (Realised at the end there is a table – ensure this is mentioned in text although I may have just missed it).

The first half of 2b is describing how the hospice is run – its not really analysis, this needs to go further if it is to be included. The last two quotes are interesting and provide insight and discuss person centred care – treating the individual. Is there more of this in the data and perhaps this section could focus on person-centred care? We often talk about it but often we don’t know how to deliver it – staff 7 seems to get it and even staff 6 is highlighting they’d not reply on a handover sheet.

2c sounds brilliant – Staff 7 quote again sounds PCC like. I’m not sure there is enough here for its own subtheme on information sharing as seems a one off point unless there’s more to it or could fit else where?

2d seems to be around decision making

6 subthemes is a lot for theme 2 – could these be synthesised together more?

3b – what is the East Cheshire model of care? A lot of this could fit with the PCC as above.

To me this is an excellently run service, and more detail about how they’re able to do this would be useful. For example, how can other services foster this environment and attitude? Perhaps goes beyond the scope of the paper but if the results could highlight this would be brilliant.

Overall less description and more synthesis and analysis.

Discussion

I would cut the second sentence. I think the introduction paragraph could be stronger. The authors found a service that used joined up care, information sharing and were delivering person-centred care and decision making at the end-of-life for people with dementia, despite a lack of funding. I would really summarise this here.

The following four paragraphs of the discussion appears to be introduction of a service apart from the odd sentence which focuses on the study results. There is little reflection of the results and placing them in context with other research – in fact no references are used in the discussion at all. I would cut all the description of the service and move to intro/method and then focus here on the results and implications.

Reviewer #2: Thank you for the opportunity to review the paper on "The East Cheshire (Teach) Dementia Programme: Increasing awareness about end-of-life care." This paper deals with the views of hospice staff and bereaved carers of people with dementia. As dementia becomes a significant healthcare issue, it is important to explore and study all aspects of this illness.

Here are several recommendations for the authors:

1. The main obstacle of the paper is that it is based on 12 interviews of less than 40 seconds each. Although it is acceptable to conduct such a study, its conclusions must be taken with caution and in the context of the place where the study was conducted.

2. You have mentioned that little research has been done on carers of people with dementia. However, there are several papers that explore medical staff and family caregivers of dementia patients in Israel. These studies were conducted on more than 80 interviewees and also examine home hospice care vs. inpatient care, views on treatment, and obstacles in implementing these care methods.

3. You have described the recruitment procedure of the interviewees in a detailed and professional manner. I would suggest adding examples of the questions they were asked.

4. With only 12 interviewees, have you reached data saturation in the data collection process?

5. You describe the process, including information sharing methods, in the Cheshire region. Is it similar to other regions in England? You also mention the Sunflower Center, and it is not clear whether this center works similarly to other day care centers. Additionally, you should elaborate more on this center and its methods of care in the introduction.

6. Regarding the barriers, you focused on the lack of resources, which is probably true. However, other studies have shown additional barriers such as a lack of knowledge and unclear regulations, among others.

7. During the discussion, you mention the H@H rapid response and discuss the differences between this and inpatient care. I believe some of this discussion, including the presentation of these two options, should be moved to the introduction.

8. In the discussion, the authors state that "Family carers whose loved ones stayed without specialist hospice care support (e.g., people remaining in their home without specialist end-of-life care or those admitted to hospital inpatient care) may be at increased risk of emotional turmoil and sorrows due to a lack of proper bereavement support." I do not think that you can conclude this from 12 interviews. You may hypothesize it, but I wouldn't draw a conclusion based solely on that.

6. PLOS authors have the option to publish the peer review history of their article (what does this mean?). If published, this will include your full peer review and any attached files.

Reviewer #1: **Yes: **Megan Armstrong

Reviewer #2: **Yes: **Rachel Nissanholtz-Gannot

---

## [Author Response · Author response to Decision Letter 0]

20 Jul 2023

Thank you for the comments/suggestions. We hope we have been able to addressed them. We have now uploaded a word document containing a table where we describe where and how in the text we have addressed the comments from reviewers.

---

## [Decision Letter · Decision Letter 1]

13 Sep 2023

PONE-D-23-14587R1Experiences of hospice dementia care: A qualitative study of bereaved carers and hospice clinicians.PLOS ONE

Dear Dr. Bosco,

Thank you for submitting your manuscript to PLOS ONE. After careful consideration, we feel that it has merit but does not fully meet PLOS ONE’s publication criteria as it currently stands. Therefore, we invite you to submit a revised version of the manuscript that addresses the points raised during the review process.

These were corrections that were raised in the previous reviews which need to be addressed. i suggest that you implement the changes as stated for a better output. 

Abstract

results

-This part of the abstract should be under methods since it is reflecting the number and type of participants that were recruited and interviewed “A total of 12 participants were interviewed from one service in the Northwest region in the UK. All were female and white British.”

Methods

Nothing was mentioned with regards to “data saturation point”. Did you at any point of the interview achieve data saturation? Please state whether or not this was achieved as part of the methods. If data saturation was not achieved during the interview, kindly state it as part of the limitation of your study.

The statement below should be part of the methods involved in recruitment of participants and should not be part of the results. Please, insert this section under methods in the appropriate location:

“Semi-structured interviews were held with 12 participants (seven clinical hospice staff and five family carers, all were female and white British) form one service in the Northwest region. Two staff were medically trained in general practice, three were nurses with one being a palliative care specialist, two were health care assistants. The interviews lasted on average 38.30 minutes.”

-Please, do include a table that will clearly indicate the THEMES, SUBTHEMES, CODES GENERATED and REPEATITION OF CODES by participants for more clarity on the data analysis and understanding of the data at a glance. This table should be in the result section and modified as

We look forward to receiving your revised manuscript.

Kind regards,

Ayi Vandi Kwaghe, D.V.M., M.V.Sc., P.G.D.E. Ph.D., MPH

Academic Editor

PLOS ONE

Journal Requirements:

Reviewers' comments:

Reviewer's Responses to Questions

**Comments to the Author**

1. If the authors have adequately addressed your comments raised in a previous round of review and you feel that this manuscript is now acceptable for publication, you may indicate that here to bypass the “Comments to the Author” section, enter your conflict of interest statement in the “Confidential to Editor” section, and submit your "Accept" recommendation.

Reviewer #1: All comments have been addressed

Reviewer #2: All comments have been addressed

2. Is the manuscript technically sound, and do the data support the conclusions?

Reviewer #1: Yes

Reviewer #2: Yes

3. Has the statistical analysis been performed appropriately and rigorously? 

Reviewer #1: N/A

Reviewer #2: Yes

4. Have the authors made all data underlying the findings in their manuscript fully available?

Reviewer #1: Yes

Reviewer #2: Yes

5. Is the manuscript presented in an intelligible fashion and written in standard English?

Reviewer #1: Yes

Reviewer #2: Yes

6. Review Comments to the Author

Reviewer #1: Thank you for the revised version. All my comments have been addressed sufficiently. One change I would suggest when proofing is to change the reflexive TA citation to codebook TA (as you do not discuss author background in data analysis section and outline a codebook). However I do not want to delay publication so suggest this is done at the proofing stage.

Reviewer #2: Thank you for the opportunity to review the manuscript on "Experiences of hospice dementia care: A qualitative study of bereaved carers and hospice clinicians". The authors conclude that patients might prefer rapid response teams for delivering hospice home care, however, this type of care should be planned. I understand that this manuscript has been changed by the authors.

My minor comment is that another limitation of the study (besides the small sample, that the authors raised) is that all participants are female and white British. Patients who are men and/or not white British might have different needs.

7. PLOS authors have the option to publish the peer review history of their article (what does this mean?). If published, this will include your full peer review and any attached files.

Reviewer #1: **Yes: **Megan Armstrong

Reviewer #2: **Yes: **Rachel Nissanholtz - Gannot

---

## [Author Response · Author response to Decision Letter 1]

18 Sep 2023

Thank you for all comments. We have now uploaded a word document containing a table where we describe where and how in the text we have addressed the comments from the editorial team and reviewers.

---

## [Editor Report · Decision Letter 2]

6 Oct 2023

Experiences of hospice dementia care: A qualitative study of bereaved carers and hospice clinicians.

PONE-D-23-14587R2

Dear Dr. Bosco,

We’re pleased to inform you that your manuscript has been judged scientifically suitable for publication and will be formally accepted for publication once it meets all outstanding technical requirements.

Kind regards,

Ayi Vandi Kwaghe, D.V.M., M.V.Sc., P.G.D.E. Ph.D., MPH

Academic Editor

PLOS ONE

---

## [Editor Report · Acceptance letter]

26 Oct 2023

PONE-D-23-14587R2 

Experiences of hospice dementia care: A qualitative study of bereaved carers and hospice clinicians. 

Dear Dr. Bosco:

I'm pleased to inform you that your manuscript has been deemed suitable for publication in PLOS ONE. Congratulations! Your manuscript is now with our production department. 

Kind regards, 

on behalf of

Dr. Ayi Vandi Kwaghe 

Academic Editor

PLOS ONE